# Comprehensive Review of Early and Late Toxicities in CAR T-Cell Therapy and Bispecific Antibody Treatments for Hematologic Malignancies

**DOI:** 10.3390/cancers17020282

**Published:** 2025-01-17

**Authors:** Ayrton Bangolo, Behzad Amoozgar, Charlene Mansour, Lili Zhang, Sarvarinder Gill, Andrew Ip, Christina Cho

**Affiliations:** 1Department of Hematology and Oncology, John Theurer Cancer Center, Hackensack University Medical Center, Hackensack, NJ 07601, USA; behzad.amoozgar@hmhn.org (B.A.); lili.zhang@hmhn.org (L.Z.); sarvarinder.gill@hmhn.org (S.G.); 2Rutgers New Jersey Medical School, Newark, NJ 07103, USA; cm1621@njms.rutgers.edu; 3Division of Lymphoma, John Theurer Cancer Center, Hackensack University Medical Center, Hackensack, NJ 07601, USA; andrew.ip@hmhn.org; 4Division of Stem Cell Transplant and Cellular Therapy, John Theurer Cancer Center, Hackensack University Medical Center, Hackensack, NJ 07601, USA; christina.cho@hmhn.org

**Keywords:** CAR-T therapy, bispecific T-cell engager therapies, hematologic malignancies, complications

## Abstract

This comprehensive review delves into the dual innovations of CAR T-cell therapy and bispecific antibody (BsAb) treatments, which have revolutionized the management of hematologic malignancies. It addresses both early and late toxicities associated with these therapies, such as cytokine release syndrome (CRS), neurotoxicity, infections, and potential secondary malignancies. This review emphasizes the importance of early detection using biomarkers and imaging and highlights intervention strategies, including immunosuppressants, corticosteroids, and cytokine inhibitors. With a focus on a multidisciplinary approach, this study underscores the need for collaboration between hematologists, oncologists, and infectious disease specialists to balance efficacy with toxicity. Emerging technologies, including targeted biologics and personalized medicine, show promise in improving patient outcomes by minimizing complications while extending survival and quality of life. This analysis serves as a critical resource for healthcare professionals managing the complexities of these advanced therapies.

## 1. Introduction

Cellular and immunotherapeutic modalities have transformed the landscape of treatment for hematologic malignancies, particularly for patients with refractory or relapsed disease. Among these therapies, CAR T-cell therapy and BsAbs have emerged as powerful tools, especially in B-cell malignancies. CAR T-cell therapy involves the genetic modification of T cells to recognize and eliminate malignant cells by targeting specific antigens, such as CD19, achieving durable remissions in some patients [1,2]. In recent years, BsAbs, including bispecific T-cell engager (BiTE) therapies, have further expanded the immunotherapy arsenal, providing a novel approach that bridges T cells and cancer cells. By binding to CD3 on T cells and a specific tumor antigen, such as CD20 or CD19, bispecific antibodies facilitate a targeted immune response against malignant cells [3,4].

Despite their efficacy, both CAR T-cell therapy and BsAbs introduce a complex array of early and delayed complications that can significantly influence patient outcomes. Early complications, such as cytokine release syndrome (CRS) and immune effector cell-associated neurotoxicity syndrome (ICANS), are driven by the acute immune activation inherent to CAR T-cell and bispecific antibody treatments. CRS, one of the most prevalent complications, results from a surge of cytokines released by activated T cells, leading to systemic inflammation that can be life-threatening if not managed effectively [5,6]. ICANS is another major complication that impacts the central nervous system, causing a spectrum of symptoms from mild confusion to severe cerebral edema. Prompt intervention, often with corticosteroids or IL-6 inhibitors like tocilizumab, is critical to mitigate these early toxicities and ensure patient safety [7,8]. Figure 1 illustrates the mechanism of activation of CRS.

In contrast, delayed complications arise from prolonged immune reconstitution and chronic immune dysregulation following CAR T-cell or bispecific therapies. A key delayed complication of allogeneic HSCT is graft-versus-host disease (GVHD), where donor T cells mount an immune response against the recipient’s tissues, leading to chronic, often debilitating symptoms [9,10]. Furthermore, CAR T-cell and bispecific antibody-treated patients face an elevated risk of cytopenias, chronic infections, and secondary malignancies due to long-term impacts on immune function. Such complications necessitate vigilant monitoring and long-term management strategies to optimize therapeutic outcomes and maintain quality of life [11,12].

The advent of these immunotherapies has introduced new complexities in the pathophysiology of treatment-related complications. Patient-specific factors, such as baseline immune function, underlying malignancy characteristics, and therapeutic approach, contribute to each individual’s risk profile. For instance, bispecific antibodies are known to induce CRS, yet they may exhibit distinct neurotoxicity profiles compared to CAR T-cell therapies [13,14]. Additionally, the risk of infections remains a significant concern, as prolonged immune suppression leaves patients vulnerable to opportunistic infections long after treatment completion. A thorough understanding of these mechanisms is critical to developing targeted interventions that minimize adverse effects while preserving the efficacy of these promising therapies [15,16].

This review aims to provide an in-depth overview of the clinical applications, complications, and management strategies associated with CAR T-cell and bispecific therapies in hematologic malignancies. By exploring current advances and challenges, we hope to inform the development of safer, more effective treatment approaches that improve patient outcomes in the long term.

## 2. Material and Methods

### 2.1. Study Design

This review synthesizes existing literature on CAR T-cell and bispecific therapies in hematologic malignancies, focusing on complications and management strategies. A narrative approach was employed to integrate findings from preclinical studies, clinical trials, and expert consensus documents. Emphasis was placed on key therapeutic agents, including CAR-T-cell therapies and bispecific T-cell engagers, their efficacy, and associated complications.

### 2.2. Data Sources

The primary databases used for literature search included PubMed, Google Scholar, and Web of Science, targeting peer-reviewed articles from 2010 to 2024. Keywords such as “CAR-T therapy complications”, “bispecific antibody toxicity”, “CRS”, and “ICANS” were used. Additionally, relevant clinical guidelines from organizations like American Society for Transplantation and Cellular Therapy (ASTCT) were included.

#### 2.2.1. Inclusion and Exclusion Criteria

Inclusion Criteria:Peer-reviewed articles discussing the clinical use of CAR-T and bispecific therapies;Studies reporting complications, including cytokine release syndrome (CRS) and immune effector cell-associated neurotoxicity syndrome (ICANS);Articles focusing on diagnostic biomarkers, imaging techniques, and management protocols.

Exclusion Criteria:Non-English language publications;Studies unrelated to hematologic malignancies or cellular therapies.

#### 2.2.2. Data Extraction

Key data points, including patient demographics, treatment regimens, complications, and outcomes, were extracted from each article. Information was categorized into early and late complications, and specific focus was given to grading systems and therapeutic interventions.

#### 2.2.3. Ethical Considerations

All included studies adhered to ethical standards approved by their respective institutional review boards. No new patient data were collected for this review.

#### 2.2.4. Analysis and Presentation

Thematic analysis was employed to synthesize data, with key findings categorized into diagnostic approaches, treatment strategies, and emerging therapies. Table 1 was created to summarize evidence systematically. Furthermore Table 2 was added with a list of abbreviations used in the manuscript.

This structured methodology ensures a comprehensive and clinically relevant overview, serving as a resource for practitioners managing complications of cellular therapies.

## 3. Indications for CAR-T and Bispecific Therapies in Hematologic Malignancies

### 3.1. B-Cell Acute Lymphoblastic Leukemia (B-ALL)

CAR-T therapy has demonstrated significant efficacy in relapsed or refractory B-cell acute lymphoblastic leukemia (B-ALL), particularly in patients under the age of 25. Tisagenlecleucel, a CD19-directed CAR-T-cell therapy, received FDA approval based on a pivotal trial where 81% of pediatric and young adult patients with relapsed or refractory B-ALL achieved a complete remission (CR) within three months of infusion, with 60% maintaining durable remissions at 12 months [17]. Additionally, bispecific antibodies such as blinatumomab, which targets both CD3 on T cells and CD19 on B cells, have shown promising outcomes in adult B-ALL patients. In the TOWER trial, blinatumomab demonstrated a CR rate of 44% in relapsed or refractory adult B-ALL patients, with a median overall survival (OS) of 7.7 months compared to 4.0 months in the standard chemotherapy group [18,19]. These therapies provide highly effective options for patients at high risk of relapse, with CAR-T offering durable responses and bispecific antibodies providing an off-the-shelf alternative for rapid disease control.

### 3.2. Multiple Myeloma (MM)

In the setting of relapsed or refractory multiple myeloma (MM), CAR-T therapies targeting B-cell maturation antigen (BCMA) have emerged as powerful treatment options. Idecabtagene vicleucel (ide-cel) and ciltacabtagene autoleucel (cilta-cel) are two BCMA-directed CAR-T therapies that have shown remarkable efficacy in patients who have exhausted other treatments, with response rates exceeding 80% in clinical trials [20]. Bispecific antibodies, such as teclistamab, which binds to BCMA on myeloma cells and CD3 on T cells, provide an off-the-shelf treatment option, especially useful for patients who may not be candidates for CAR-T therapy due to manufacturing time or prior lymphodepletion requirements [21,22].

### 3.3. Chronic Lymphocytic Leukemia (CLL)

While CAR-T therapy is less commonly utilized in chronic lymphocytic leukemia (CLL) due to the disease’s generally indolent nature, it has demonstrated efficacy in cases of Richter’s transformation, where CLL progresses to an aggressive lymphoma. In a multicenter retrospective study evaluating anti-CD19 CAR-T therapy for Richter’s transformation, an overall response rate (ORR) of 63% was reported, with 46% achieving complete remission (CR). Additionally, the median overall survival (OS) for responding patients was 8.5 months, highlighting its potential as a therapeutic option for this aggressive condition [23,24]. For standard CLL, bispecific antibodies are being investigated as promising alternatives. In a phase I trial, glofitamab, a bispecific antibody targeting CD20 and CD3, achieved a CR rate of 39% and an ORR of 64% in relapsed/refractory B-cell lymphomas, suggesting its potential utility in CLL patients who have relapsed after CAR-T therapy or are unsuitable candidates for intensive treatments [25].

### 3.4. Mantle Cell Lymphoma (MCL)

Mantle cell lymphoma (MCL) has become a significant area of focus for CAR-T therapies. Brexucabtagene autoleucel, a CD19-directed CAR-T therapy, has demonstrated substantial response rates in patients with relapsed or refractory MCL, providing a durable treatment option for those who have failed other therapies [2]. Axicabtagene ciloleucel has also demonstrated continued durable responses in relapsed and refractory indolent non-Hodgkin lymphoma [26]. Given the aggressive nature of MCL and frequent relapses, CAR-T therapy is particularly beneficial. In addition, bispecific antibodies like glofitamab are currently under investigation as a complementary option to CAR-T for MCL [27].

### 3.5. Diffuse Large B-Cell Lymphoma (DLBCL)

Diffuse large B-cell lymphoma (DLBCL) is among the most common hematologic malignancies treated with CAR-T therapy. FDA-approved CAR-T products such as axicabtagene ciloleucel, tisagenlecleucel, and lisocabtagene maraleucel have demonstrated effectiveness in patients with relapsed or refractory disease, achieving response rates of 50–80% depending on the specific product [28]. In addition, bispecific antibodies like epcoritamab, targeting CD20 and CD3, are under investigation in DLBCL and may provide a viable option for patients who cannot tolerate the toxicities associated with CAR-T therapy [29].

### 3.6. Follicular Lymphoma (FL)

Follicular lymphoma (FL), although typically indolent, can progress to a more aggressive form requiring novel therapies. CAR-T therapy has shown efficacy in relapsed or refractory FL, with CD19-directed products such as axicabtagene ciloleucel achieving an overall response rate (ORR) of 94% and a complete response (CR) rate of 79% in clinical trials. Durable remissions have been reported, with a median progression-free survival (PFS) of 39.6 months in responders [26]. For patients seeking alternatives to CAR-T therapy, bispecific antibodies such as mosunetuzumab, which targets CD20 on B cells and CD3 on T cells, demonstrated an ORR of 80% and a CR rate of 60% in heavily pretreated FL patients in early-phase trials. These therapies offer a less intensive treatment option with manageable side effects, including low-grade cytokine release syndrome (CRS) in approximately 44% of patients, most of which were Grade 1 or 2 [30].

## 4. Early Complications of CAR-T Therapy: Diagnostic Workup and Management

### 4.1. Introduction to Early Complications

CAR-T-cell therapy has emerged as a powerful treatment for various hematologic malignancies, yet it is often accompanied by significant early complications. These adverse effects, predominantly CRS and ICANS, arise from the intense immune activation associated with CAR-T cells targeting tumor antigens. Understanding the diagnostic workup for these complications is essential for ensuring timely and effective management, ultimately improving patient outcomes [31,32,33].

### 4.2. Cytokine Release Syndrome (CRS) Overview

CRS is the most frequent and well-documented early complication of CAR-T therapy, occurring in a majority of patients within days of infusion. It occurs in 50–90% of patients depending on the CAR construct and target antigen, such as CD19 or BCMA. Severe CRS (Grade ≥ 3) has been reported in approximately 10–20% of cases, often requiring interventions such as vasopressors or mechanical ventilation. For instance, in the pivotal ZUMA-1 trial evaluating axicabtagene ciloleucel, CRS was observed in 93% of patients, with Grade ≥ 3 CRS in 13%. CRS results from a robust release of inflammatory cytokines—most notably interleukin-6 (IL-6)—due to CAR-T-cell proliferation and activation [7,11].

CRS symptoms vary widely, ranging from mild flu-like symptoms to severe systemic inflammation, which can lead to hypotension, high fever, and multi-organ failure. The severity of CRS is categorized into four grades, each guiding specific treatment strategies [7,34].

### 4.3. Diagnostic Workup for CRS in CAR-T Therapy

The diagnostic process for CRS involves grading the severity based on clinical presentation and laboratory markers. For patients showing signs of CRS, regular monitoring of vital signs and inflammatory markers, such as C-reactive protein (CRP) and ferritin, is critical. Elevated CRP and ferritin levels correlate with increased inflammation and can indicate the intensity of CRS. For higher-grade CRS (Grades 3 and 4), additional diagnostic measures, including echocardiography and blood gas analysis, may be warranted to assess organ function and guide therapy [5,35,36]. Imaging and more invasive testing may be required in severe cases to rule out other causes of systemic inflammation.

### 4.4. Management of CRS Based on Diagnostic Findings

Management strategies for CRS are tailored according to the CRS grade. Mild cases (Grade 1) typically require supportive care, including antipyretics and intravenous fluids. However, for moderate to severe cases (Grades 2–4), treatment with tocilizumab—a monoclonal antibody targeting IL-6 receptors—is recommended and can be administered every 8 h if symptoms persist. In cases where tocilizumab alone is insufficient, corticosteroids, such as dexamethasone, may be added to control the inflammatory response. Severe cases (Grade 4) often necessitate transfer to the intensive care unit for close monitoring and advanced supportive measures, such as vasopressors or mechanical ventilation if necessary [23,37,38].

### 4.5. Immune Effector Cell-Associated Neurotoxicity Syndrome (ICANS) Overview

ICANS (immune effector cell-associated neurotoxicity syndrome) is a significant and potentially severe early complication of CAR-T-cell therapy, often occurring following or concurrently with cytokine release syndrome (CRS). Epidemiological studies have shown that the incidence of ICANS ranges from 20% to 64% in patients undergoing CAR-T-cell therapy, with higher-grade neurotoxicity (Grade ≥ 3) reported in 10–28% of cases, depending on the CAR-T construct and underlying malignancy. For example, ICANS was observed in 64% of patients treated with axicabtagene ciloleucel for large B-cell lymphoma, with severe cases in 28% [39,40,41].

This neurotoxic complication is characterized by a spectrum of symptoms, including confusion, aphasia, tremors, and, in severe cases, seizures, status epilepticus, and cerebral edema. These symptoms can escalate rapidly, underscoring the need for immediate diagnosis and intervention. The pathophysiology of ICANS is complex and remains incompletely understood. Current evidence suggests it is mediated by both direct effects of CAR-T cells trafficking to the central nervous system (CNS) and secondary inflammatory responses, including elevated cytokines such as IL-6, IFN-γ, and TNF-α, which disrupt the blood–brain barrier [39,40,41].

ICANS typically manifests within a few days of CRS onset, often mirroring its severity. Prompt diagnosis relies on standardized grading tools such as the Immune Effector Cell-Associated Encephalopathy (ICE) score, which evaluates the patient’s orientation, handwriting ability, and command-following, among other parameters. Early recognition and grading are critical to implementing timely interventions, which may include corticosteroids and supportive neurocritical care [39,40,41].

## 5. Late Complications of CAR-T Therapy: Diagnostic Workup and Management

### 5.1. Introduction to Late Complications in CART-Therapy

While CAR-T therapy has demonstrated significant efficacy in treating hematologic malignancies, it is associated with various late complications, often emerging weeks to months post-treatment. These complications can include prolonged cytopenias, infections, B-cell aplasia, and, in some cases, secondary malignancies. Understanding these risks and implementing effective diagnostic workup and management strategies are essential for optimizing long-term patient outcomes [42,43,44].

### 5.2. Prolonged Cytopenias in CART-Therapy

Prolonged cytopenias are among the most common late complications of CAR-T therapy, often resulting from the lymphodepletion regimen combined with CAR-T-cell activity. These cytopenias—manifesting as persistent neutropenia, thrombocytopenia, and anemia—are reported in approximately 30–40% of patients, with rates varying based on the CAR construct, disease type, and patient characteristics. Persistent neutropenia significantly elevates the risk of opportunistic infections, while thrombocytopenia increases bleeding risks, and anemia contributes to fatigue and reduced quality of life. Studies have shown that Grade ≥ 3 neutropenia persists beyond 30 days in 15–20% of patients, with thrombocytopenia observed in a similar proportion. In a large cohort study, 37% of patients required prolonged transfusion support for anemia and thrombocytopenia more than one month post-infusion. Persistent cytopenias extending beyond 90 days occur in approximately 10–15% of patients, necessitating further investigation to rule out bone marrow failure or underlying myelodysplasia [45,46,47].

The diagnostic approach includes regular complete blood counts to monitor for cytopenias and bone marrow biopsies if cytopenias persist beyond three months to rule out bone marrow failure or underlying myelodysplasia. Management typically involves growth factor support (such as G-CSF for neutropenia), transfusions, and, in some cases, immunosuppressive therapies if an autoimmune etiology is suspected [45,46,47].

### 5.3. Infections and Opportunistic Pathogens in CART-Therapy

Due to the immunosuppressive effects of CAR-T therapy and prolonged cytopenias, patients are at increased risk of infections, particularly from opportunistic pathogens. Early in the course of therapy, bacterial and viral respiratory infections are most common, occurring in approximately 40–50% of patients. As time progresses, viral reactivations, including herpes simplex virus (HSV) and cytomegalovirus (CMV), occur in 20–30% of patients, while fungal infections, such as invasive aspergillosis, are observed in 10–15% of cases [30,48,49].

Diagnostic workup includes routine blood cultures, viral PCR testing, and radiological imaging to identify infection sources. Preventive measures such as prophylactic antivirals, antifungals, and antibiotics, along with immunoglobulin replacement therapy, are recommended to protect against recurrent infections [30,48,49].

### 5.4. B-Cell Aplasia and Hypogammaglobulinemia in CART-Therapy

CAR-T therapy, especially when targeting CD19, can lead to prolonged B-cell aplasia due to the depletion of both malignant and normal B cells. This condition results in hypogammaglobulinemia, which increases susceptibility to recurrent sinopulmonary infections. Diagnostic workup involves monitoring serum immunoglobulin levels and B-cell counts. Management typically includes regular immunoglobulin replacement therapy to maintain adequate IgG levels and prevent infections until B-cell recovery, which can take months to years in some patients [50,51,52].

### 5.5. Secondary Malignancies in CART-Therapy

Secondary malignancies following CAR-T therapy, though rare, are a significant consideration in long-term patient management. These malignancies may result from prior chemotherapy, radiation, or insertional mutagenesis associated with the genetic modification of T cells. Additionally, pre-existing clonal hematopoiesis or extensive prior treatment history can increase this risk. Routine surveillance and early detection are essential to mitigate morbidity and improve outcomes [53,54,55]. Authors recommended surveillance modalities include the following:Imaging: Imaging should be considered, such as CT or PET-CT scans, to monitor for solid tumors or hematologic malignancies based on patients’ complaints and on a case-to-case basis. MRI can be considered for patients requiring reduced radiation exposure;Hematologic Assessments: Routine complete blood counts (CBC) with differential and peripheral smear evaluations to identify early signs of hematologic malignancies. Molecular testing and bone marrow biopsy may be warranted for unexplained cytopenias or other abnormal findings;Monitoring for Clonal Hematopoiesis: Regular assessment for clonal hematopoiesis of indeterminate potential (CHIP) using next-generation sequencing (NGS), particularly in patients with significant genotoxic exposure history;Standard Cancer Screening: Adherence to age- and gender-appropriate cancer screening protocols, such as mammography, colonoscopy, or low-dose CT scans for lung cancer in high-risk individuals.

### 5.6. Graft-Versus-Host Disease (GVHD)

In patients who have undergone prior allogeneic stem cell transplants, CAR-T therapy may trigger or exacerbate graft-versus-host disease (GVHD). GVHD can affect multiple organs, presenting with symptoms like skin rash, liver dysfunction, and gastrointestinal disturbances. Diagnostic workup includes biopsy of affected tissues, liver function tests, and endoscopy if gastrointestinal involvement is suspected. Management typically involves corticosteroids and additional immunosuppressive agents, with more severe cases requiring specialized treatment, such as ruxolitinib for steroid-refractory GVHD [56,57,58,59].

### 5.7. Neurological Complications

While acute neurotoxicity, ICANS, is well recognized, delayed neurological complications can also occur after CAR-T therapy. These can include cognitive impairments, mood changes, and, in rare cases, chronic seizures. Neurological assessment through regular monitoring, magnetic resonance imaging (MRI), and electroencephalogram (EEG) can help in diagnosing late-onset neurotoxicity. Management may involve symptomatic treatments, anti-epileptic medications, and supportive neuropsychiatric care for persistent cognitive symptoms [60,61,62].

### 5.8. Management of Late Complications

Management of late CAR-T complications is multifaceted and requires a long-term, multidisciplinary approach. Patients benefit from routine follow-ups with hematologists, infectious disease specialists, and neurologists to monitor and manage ongoing issues. For infections, prophylactic antimicrobials and immunoglobulin replacement are crucial. For cytopenias, growth factors and transfusions may be necessary. Additionally, patients should undergo routine surveillance for secondary malignancies and ongoing neuropsychological evaluations. Advances in CAR-T technology aim to reduce these late toxicities, with research focused on refining CAR designs and identifying biomarkers to predict long-term outcomes [43,48,63,64,65].

## 6. Early Complications of Bispecific Antibodies: Diagnostic Workup and Management

### 6.1. Introduction to Bispecific Antibodies and Early Complications

BsAbs, including BiTEs, have emerged as effective immunotherapy agents in hematologic malignancies. These agents function by simultaneously binding a tumor antigen, such as CD19 or BCMA, and a T-cell marker, like CD3, facilitating T-cell-mediated lysis of cancer cells. BsAbs have demonstrated significant clinical efficacy, with agents such as blinatumomab achieving an overall response rate (ORR) of 44% and a complete remission (CR) rate of 33% in relapsed/refractory B-cell acute lymphoblastic leukemia (B-ALL). Similarly, teclistamab, a BCMA-targeting BsAb for relapsed/refractory multiple myeloma, demonstrated an ORR of 63% and a CR rate of 39% in heavily pretreated patients [66,67].

Despite their efficacy, BsAbs are associated with early complications, predominantly cytokine release syndrome (CRS) and neurotoxicity. CRS has been reported in up to 70% of patients receiving BsAbs, with severe (Grade ≥ 3) cases occurring in approximately 5–10%. Neurotoxicity, while less common, affects 10–15% of patients and can range from mild confusion to seizures or cerebral edema [68]. These complications, similar to those observed with CAR-T-cell therapies, necessitate precise diagnostic and management protocols to ensure safety and optimize therapeutic outcomes.

### 6.2. Cytokine Release Syndrome (CRS) in BsAbs

CRS is a prevalent and clinically significant complication of BsAb therapy, occurring as a result of rapid cytokine release when T cells are activated upon engagement with tumor cells. Recent studies have reported the incidence of CRS in BsAb therapy to range between 60% and 80%, with severe cases (Grade ≥ 3) occurring in approximately 5–10% of patients, depending on the specific BsAb and patient population studied. For example, teclistamab, a BCMA-targeting BsAb for relapsed/refractory multiple myeloma, demonstrated an overall CRS rate of 72%, with Grade 3 CRS in <1% of cases [36,39,67].

CRS in BsAb therapy is typically milder compared to that seen with CAR-T-cell therapies, yet it still necessitates vigilant monitoring. Mild symptoms often include fever, fatigue, and myalgias, while severe manifestations can progress to hypotension requiring vasopressors, hypoxia necessitating oxygen supplementation, and multi-organ dysfunction [36,39,67].

To facilitate early detection and grading of CRS, routine monitoring of inflammatory biomarkers is essential. Elevations in C-reactive protein (CRP) and ferritin are commonly observed and serve as surrogate markers for cytokine-mediated inflammation. Additionally, interleukin-6 (IL-6) levels have been identified as a key biomarker, with studies correlating IL-6 elevations with CRS severity. For patients presenting with respiratory symptoms, imaging such as chest X-rays or computed tomography (CT) scans may aid in identifying complications such as pulmonary edema or fluid overload, ensuring timely intervention [36,39,67,69,70,71,72,73].

### 6.3. Diagnostic Workup for CRS in BsAb Therapy

The diagnostic approach to CRS in BsAb therapy emphasizes regular clinical assessment and laboratory evaluations. For patients undergoing BsAb treatment, baseline levels of CRP and ferritin are established, and these markers are subsequently monitored to track inflammation. In cases where symptoms indicate severe CRS, additional tests like liver function tests, lactate dehydrogenase, and renal function panels help assess the impact on organ systems. Imaging, such as chest X-rays, may be necessary for patients presenting with respiratory symptoms, allowing for the identification of any fluid overload or pulmonary edema [72,73,74].

### 6.4. Management of CRS in BsAb Therapy

Management of CRS associated with BsAbs is tiered based on severity. For low-grade CRS, symptomatic treatment such as antipyretics and hydration is typically sufficient. In moderate to severe cases, tocilizumab, an IL-6 receptor antagonist, is administered, with corticosteroids added if symptoms persist or worsen. Patients with severe CRS may require intensive care monitoring, where vasopressors and respiratory support can be provided as necessary [75,76].

### 6.5. Neurotoxicity and ICANS in Bispecific Antibodies

Neurotoxicity is a less common but potentially severe early complication of bispecific antibody (BsAb) therapy, with an incidence ranging between 10 and 15% depending on the BsAb agent and patient population. This complication presents with a spectrum of symptoms, including mild confusion, tremors, aphasia, and, in severe cases, seizures, status epilepticus, and cerebral edema. Although neurotoxicity in BsAb therapy is less frequent and often milder compared to ICANS (immune effector cell-associated neurotoxicity syndrome) observed with CAR-T-cell therapy, its occurrence necessitates prompt diagnosis and management to prevent irreversible neurological damage [67,75,76,77].

Unlike CRS, neurotoxicity with BsAb therapy typically emerges later, often following or coinciding with CRS resolution. This temporal distinction requires vigilant monitoring of patients even after CRS symptoms subside. The pathophysiology of neurotoxicity in BsAb therapy is not fully elucidated but may involve inflammatory cytokine-mediated blood–brain barrier disruption and immune effector cell trafficking to the central nervous system (CNS). Elevated levels of cytokines such as IL-6 and TNF-α are commonly observed during episodes of neurotoxicity [67,75,76,77].

The Immune Effector Cell-Associated Encephalopathy (ICE) score is widely employed for assessing neurological function in patients receiving BsAb therapy. This tool evaluates cognitive and motor functions, including orientation, handwriting, and ability to follow commands, allowing for standardized grading of neurotoxicity severity. Early and accurate grading is critical for implementing timely interventions, such as corticosteroids or supportive neurocritical care, to mitigate long-term sequelae [67,75,76,77].

### 6.6. Diagnostic Workup for Neurotoxicity

Early diagnosis of neurotoxicity involves frequent neurological assessments using tools like the ICE score, as well as EEG monitoring in cases with severe or persistent symptoms. Imaging studies such as MRI are recommended for patients showing acute neurotoxic symptoms to rule out alternative causes like infections or brain edema. Serum biomarker analysis, focusing on inflammatory markers, supports clinical findings and can indicate the severity of the immune response involved in neurotoxicity [75,76,77].

### 6.7. Management of Neurotoxicity in BsAb Therapy

Management of neurotoxicity primarily involves corticosteroids, such as dexamethasone, which are administered based on the severity of symptoms. In patients exhibiting cerebral edema, osmotic therapy may be used to control intracranial pressure, and anti-epileptic drugs are introduced for seizure management. Supportive care with neuropsychiatric evaluation is critical for persistent symptoms. Severe neurotoxicity cases may require intensive care unit (ICU)-level care with continuous monitoring and neuroprotective strategies to mitigate long-term sequelae [75,76,77,78,79].

### 6.8. Future Directions in the Management of Early Complications in BsAbs

Ongoing research aims to refine diagnostic criteria and develop predictive biomarkers to anticipate early complications from BsAb therapy, allowing for preemptive intervention. Innovations include developing next-generation BsAbs that selectively modulate immune activation, potentially reducing the incidence and severity of CRS and neurotoxicity. Clinical trials continue to assess various BsAbs, with an emphasis on optimizing safety profiles through tailored dosing regimens and combination therapies that mitigate immune-related side effects while preserving therapeutic efficacy [75,80,81].

## 7. Late Complications of Bispecific Antibody Therapy: Diagnostic Workup and Management

### 7.1. Introduction to Late Complications in BsAb Therapy

BsAbs have become increasingly prominent in treating hematologic malignancies due to their ability to engage immune cells directly with tumor cells. However, late complications associated with BsAb therapies have emerged, which include prolonged cytopenias, B-cell aplasia, infections, and, occasionally, the development of secondary malignancies. Recognizing and managing these late-onset adverse effects is essential for optimizing patient outcomes and extending the benefits of BsAb therapy [82,83].

### 7.2. Prolonged Cytopenias in BsAb Therapy

Prolonged cytopenias are a recognized late complication of BsAb therapy, occurring in a significant proportion of patients due to sustained T-cell activation and immune system stress. Recent studies report the incidence of prolonged cytopenias, including neutropenia, thrombocytopenia, and anemia, in approximately 30–50% of patients receiving BsAbs, depending on the agent and treatment setting. These cytopenias can persist for weeks to months following treatment, increasing the risk of infections and bleeding complications, particularly in patients with pre-existing bone marrow compromise [67,75,82,83,84].

The diagnostic approach to prolonged cytopenias includes regular monitoring of complete blood counts (CBCs) to detect trends over time and identify the need for intervention. For cytopenias persisting beyond three months, bone marrow biopsies are recommended to evaluate for underlying marrow suppression, hypocellularity, or secondary myelodysplastic syndrome (MDS). These investigations are critical for ruling out disease progression or treatment-related marrow toxicity [82,83,84].

Management of prolonged cytopenias involves a combination of supportive care measures. Granulocyte colony-stimulating factor (G-CSF) is often employed to stimulate neutrophil recovery in cases of prolonged neutropenia, reducing infection risk. For thrombocytopenia and anemia, transfusions of platelets and packed red blood cells are used to manage symptoms and prevent complications. Recent clinical data suggest that individualized care plans, including dose adjustments or treatment delays, may further mitigate the impact of cytopenias while preserving therapeutic efficacy [82,83,84].

### 7.3. Infections and Opportunistic Pathogens in BsAb Therapy

Due to the immunosuppressive nature of BsAb therapies, patients are at increased risk for infections, particularly with opportunistic pathogens. Late-onset infections often involve viral reactivations, such as cytomegalovirus and herpes simplex, as well as fungal infections. A systematic review of MM patients reported that 56% of patients receiving BsAbs experienced infections of any grade, with 24% encountering grade ≥ 3 infections. Notably, patients treated with BCMA-targeted BsAbs had higher rates of severe infections compared to those receiving non-BCMA BsAbs [85].

Diagnostic workup for infections includes routine viral PCR testing, cultures, and imaging such as chest radiographs or computed tomography (CT) scans to detect respiratory or systemic infections. Management strategies involve antimicrobial prophylaxis, immunoglobulin replacement for hypogammaglobulinemia, and close monitoring for signs of infection during and after therapy [84,85,86].

### 7.4. B-Cell Aplasia and Hypogammaglobulinemia in BsAb Therapy

BsAbs targeting CD19 or CD20 can result in prolonged B-cell depletion, leading to hypogammaglobulinemia and susceptibility to bacterial infections. The diagnostic workup for B-cell aplasia includes monitoring B-cell counts and serum immunoglobulin levels. Patients with persistent B-cell aplasia may require immunoglobulin replacement therapy to maintain protective IgG levels and reduce infection risk. Management involves regular assessments and periodic intravenous immunoglobulin (IVIG) infusions, especially if patients exhibit recurrent infections [82,83,84,85,86,87].

### 7.5. Secondary Malignancies in BsAb Therapy

There is no direct evidence linking BsAb therapies to an increased risk of secondary malignancies. Rare cases of secondary cancers reported in patients undergoing BsAb therapy are likely attributable to factors such as prior treatments, genetic predispositions, or the natural progression of the underlying disease [13,82].

Given the lack of direct causative evidence, the need for routine surveillance for secondary malignancies in patients receiving BsAbs should be individualized. When surveillance is deemed appropriate, modalities may include the following:Routine Imaging: Consider periodic CT or PET scans in patients with high-risk features or a history of prior malignancies;Hematologic Assessments: Regular complete blood counts (CBC) and peripheral blood smears to monitor for unexpected hematologic abnormalities;Biopsies: For cases of suspected transformation or unexplained lesions detected on imaging.

Management of secondary malignancies, if identified, depends on the specific diagnosis and may include chemotherapy, radiation, or surgical interventions as necessary. While no clear link exists between BsAbs and malignancies, a personalized approach to monitoring remains essential to ensure patient safety and optimal outcomes [13,82].

### 7.6. Chronic Inflammatory and Autoimmune Reactions

Late-stage BsAb therapy can sometimes trigger chronic inflammatory or autoimmune reactions due to sustained T-cell activation. These reactions may present as autoimmune cytopenias, such as hemolytic anemia or thrombocytopenia, and may persist long after the completion of therapy. Diagnostic workup includes antibody testing, direct antiglobulin testing, and assessments for other autoimmune markers. Management of these reactions typically involves immunosuppressive therapies such as corticosteroids or, in severe cases, additional immunomodulatory agents [88,89,90].

### 7.7. Neurotoxicity

Although neurotoxicity is more common as an early complication, late-onset neurological symptoms can also occur with BsAb therapy. Patients may experience cognitive impairments, mood disorders, or chronic headaches. For these patients, regular neurological exams, imaging with MRI, and electroencephalograms (EEGs) may be necessary to assess for ongoing neurotoxicity. Symptomatic management includes the use of anti-epileptics for seizures and neuropsychiatric support for cognitive or mood-related symptoms [13,91,92,93]. Despite the comorbidity associated with neurotoxicity, non-relapse mortality is higher due to infections [94].

### 7.8. Management and Future Directions

Management of late complications from BsAbs necessitates a multidisciplinary approach involving hematologists, infectious disease specialists, and neurologists. Late complications such as prolonged cytopenias occur in approximately 20–30% of patients, while hypogammaglobulinemia is observed in up to 60% of those treated with BsAbs targeting CD19 or BCMA. To mitigate these risks, routine laboratory testing (e.g., complete blood counts, serum immunoglobulin levels), imaging, and supportive therapies such as intravenous immunoglobulin (IVIG) infusions are critical.

Future developments in BsAb design aim to reduce immune-related toxicities, such as cytokine release syndrome (CRS), which occurs in 30–70% of cases, with severe CRS (Grade ≥ 3) reported in approximately 5–10%. Innovations include modifying target specificity to minimize off-tumor effects and incorporating safety switches to regulate immune activation. Biomarker research is ongoing, focusing on identifying predictive markers for long-term toxicities and patient-specific risk stratification. These advancements hold promise for enhancing the safety and efficacy profiles of BsAb therapies, ultimately improving long-term outcomes for patients with hematologic malignancies [65,95,96,97,98,99].

## 8. Conclusions

CAR-T-cell therapy and bispecific antibodies have reshaped the therapeutic landscape for hematologic malignancies, providing new hope for patients with otherwise refractory or relapsed disease. However, these therapies come with a range of early and late complications that require meticulous diagnostic workup and individualized management strategies to optimize patient outcomes. Early complications such as cytokine release syndrome (CRS) and neurotoxicity demand timely intervention with protocols adapted to the severity of symptoms, utilizing tools like the ICE score for neurotoxicity and biomarkers for inflammatory response to enable prompt diagnosis and treatment. Similarly, for bispecific antibodies, early complications mirror those seen with CAR-T therapy, necessitating similar diagnostic vigilance and therapeutic strategies tailored to manage these unique toxicities effectively.

Late complications associated with both CAR-T therapy and bispecific antibodies, including prolonged cytopenias, infections, and B-cell aplasia, highlight the need for long-term monitoring and multidisciplinary care. The immunosuppressive nature of these treatments predisposes patients to opportunistic infections and, in some cases, secondary malignancies, underscoring the importance of routine follow-up with laboratory testing, imaging, and immunoglobulin replacement therapy when appropriate. For bispecific antibodies, late complications are managed through continued surveillance and supportive care, with an emphasis on mitigating autoimmune reactions and neurotoxicity that may emerge as the treatment progresses. As we refine the understanding of these late effects, personalized approaches that predict and prevent complications based on individual patient risk profiles will be essential.

Looking forward, advancements in CAR-T and bispecific antibody technologies aim to minimize these complications by enhancing the precision and safety of immune activation. Research focused on identifying predictive biomarkers, optimizing dosing regimens, and developing next-generation constructs with reduced toxicity is paving the way for safer, more effective therapies. Through an ongoing commitment to understanding and addressing the full spectrum of complications, clinicians can continue to improve the quality of life and survival outcomes for patients undergoing these transformative treatments in hematologic oncology.

## Figures and Tables

**Figure 1 cancers-17-00282-f001:**
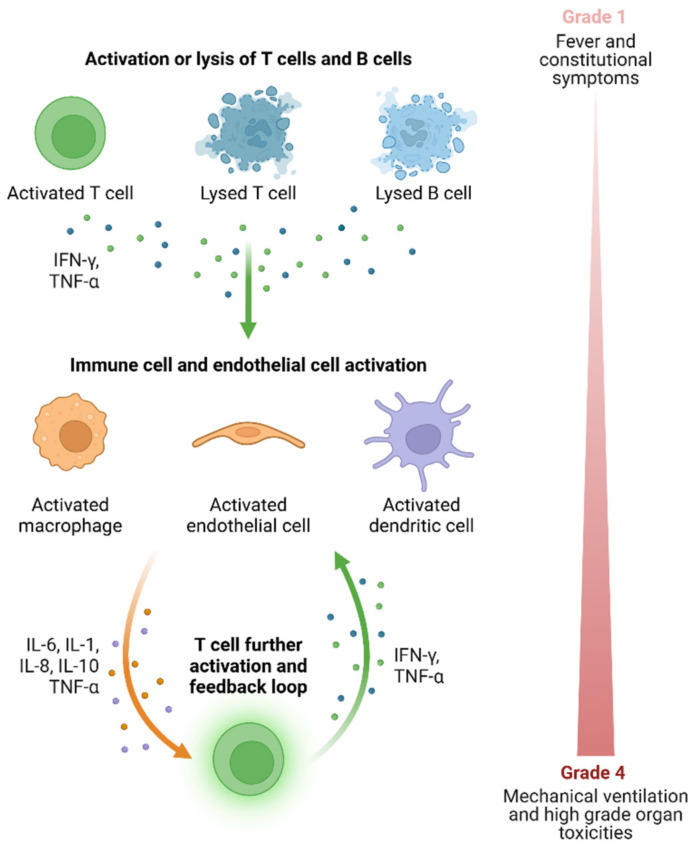
Mechanism of activation of Cytokine Release Syndrome.

**Table 1 cancers-17-00282-t001:** Overview of early and late complications associated with CAR-T and bispecific antibody therapies in hematologic malignancies.

Complication	Type	Diagnosis	Therapy Type	Affected System	Hematologic Malignancy	Management	Specific Agents	Indications	FDA Approval Status	Timing	Incidence	Severity
Cytokine Release Syndrome (CRS)	Early	Vital signs, CRP, ferritin, IL-6 monitoring, imaging for severe cases	CAR-T, Bispecific Antibodies	Immune	ALL, DLBCL, MCL, MM	Supportive care for mild cases; tocilizumab and corticosteroids for severe cases	Tocilizumab, corticosteroids	CD19-directed CAR-T therapies, BsAbs	Approved for CAR-T therapies	Within days post-infusion	Common (70–90%)	Can be severe (Grades 1–4)
ICANS	Early	ICE score, neurological exams, EEG, MRI	CAR-T	Neurological	ALL, DLBCL, MCL	Corticosteroids, anti-epileptics, supportive neuropsychiatric care	Dexamethasone	CD19-directed CAR-T therapies	Approved for CAR-T therapies	Days post-CRS	Common (30–50%)	Can be severe (Grades 1–4)
Neurotoxicity	Early	ICE score, EEG, MRI	Bispecific Antibodies	Neurological	DLBCL, MM	Corticosteroids, anti-epileptic drugs, supportive care	Dexamethasone	CD20- and BCMA-directed BsAbs	Investigational	Days post-infusion	Common (20–40%)	Can be severe (Grades 1–4)
Prolonged Cytopenias	Late	CBC, bone marrow biopsy for cases > 3 months	CAR-T, Bispecific Antibodies	Hematologic	ALL, DLBCL, MM	Growth factors (e.g., G-CSF), transfusions, immunosuppressives if autoimmune	G-CSF, transfusions	CD19-directed CAR-T, BCMA-directed	Approved for CAR-T therapies	Weeks to months post-infusion	Common (20–40%)	Variable
Infections	Late	Blood cultures, viral PCR, imaging	CAR-T, Bispecific Antibodies	Immune, Respiratory	ALL, DLBCL, MCL, CLL, MM	Antimicrobial prophylaxis, immunoglobulin replacement	IVIG, antimicrobials	CD19- and BCMA-directed CAR-T, BsAbs	Approved for CAR-T therapies	Weeks to months post-infusion	Common	Variable
B-cell Aplasia and Hypogammaglobulinemia	Late	B-cell counts, serum immunoglobulin levels	CAR-T, Bispecific Antibodies	Immune	ALL, DLBCL, FL	Immunoglobulin replacement therapy	IVIG	CD19-directed CAR-T therapies	Approved for CAR-T therapies	Months post-infusion	Common	Mild to moderate
Secondary Malignancies	Late	Routine imaging, hematologic assessments	CAR-T	Systemic	ALL, DLBCL, MM	Dependent on type; chemotherapy, radiation	Depends on malignancy	Post-CAR-T therapy complications	Investigational	Months to years post-infusion	Rare	Can be severe
Graft-versus-Host Disease (GVHD)	Delayed	Biopsy, liver function tests, endoscopy	CAR-T (Allogeneic)	Multi-organ	ALL, DLBCL	Corticosteroids, immunosuppressants, ruxolitinib	Ruxolitinib, corticosteroids	Post-allogeneic CAR-T therapy	Approved for steroid-refractory GVHD	Weeks to months post-infusion	Variable	Can be severe
Chronic Neurotoxicity	Late	Neurological exams, MRI, EEG	CAR-T, Bispecific Antibodies	Neurological	ALL, DLBCL, MM	Symptomatic management, anti-epileptics, neuropsychiatric support	Anti-epileptics, neuropsychiatric care	CD19- and BCMA-directed therapies	Investigational	Months post-infusion	Variable	Variable
Chronic Inflammatory and Autoimmune Reactions	Late	Antibody testing, direct antiglobulin tests, autoimmune markers	Bispecific Antibodies	Multi-organ	DLBCL, CLL	Corticosteroids, additional immunomodulatory agents	Corticosteroids, immunomodulators	Post-BsAb therapy	Investigational	Weeks to months post-infusion	Variable	Mild to moderate

**Table 2 cancers-17-00282-t002:** Glossary of common abbreviations in cellular therapies.

Abbreviation	Full Term
B-ALL	B-Cell Acute Lymphoblastic Leukemia
BCMA	B-Cell Maturation Antigen
BiTE	Bispecific T-Cell Engager
BsAb	Bispecific Antibody
CAR-T	Chimeric Antigen Receptor T-Cell
CLL	Chronic Lymphocytic Leukemia
CR	Complete Remission
CRP	C-Reactive Protein
CRS	Cytokine Release Syndrome
CT	Computed Tomography
DLBCL	Diffuse Large B-Cell Lymphoma
EEG	Electroencephalogram
FL	Follicular Lymphoma
FDA	Food and Drug Administration
GVHD	Graft-versus-Host Disease
G-CSF	Granulocyte Colony-Stimulating Factor
HSCT	Hematopoietic Stem Cell Transplantation
ICANS	Immune Effector Cell-Associated Neurotoxicity Syndrome
ICE Score	Immune Effector Cell-Associated Encephalopathy Score
IgG	Immunoglobulin G
IL-6	Interleukin-6
IVIG	Intravenous Immunoglobulin
MCL	Mantle Cell Lymphoma
MM	Multiple Myeloma
MRI	Magnetic Resonance Imaging
ORR	Overall Response Rate
OS	Overall Survival
PCR	Polymerase Chain Reaction
PFS	Progression-Free Survival
TME	Tumor Microenvironment
TKI	Tyrosine Kinase Inhibitor

## Data Availability

The complete datasets used and/or analyzed during this study are available from the corresponding author upon request. Requests can be made through the corresponding author (Ayrton Bangolo; Email: ayrton.bangolo@hmhn.org).

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
