# Peer review of "Comprehensive Review of Early and Late Toxicities in CAR T-Cell Therapy and Bispecific Antibody Treatments for Hematologic Malignancies"

_cancers, 2025, doi:10.3390/cancers17020282_

Round 1

Reviewer 1 Report (Previous Reviewer 1)

Comments and Suggestions for Authors

I carefully reviewed the revised manuscript by Ayrton Bangolo et al., titled "Comprehensive Review of Early and Late Toxicities in CAR T-Cell Therapy 2 and Bispecific Antibody Treatments for Hematologic Malignancies." I would like to commend the authors for their meticulous attention to the initial feedback. The inclusion of the "Materials and Methods" section, quantitative data, and visual aids such as figure and table has significantly enhanced the manuscript's clarity and comprehensiveness. These updates have improved the scientific rigor and overall readability of the article.

The manuscript is now in a strong position, but a few minor issues remain to be addressed before publication:

Minor Comments:

  1. References in Lines 254 and 261. The formatting of references on these lines requires correction.
  2. Paragraph Numbering. While the "Materials and Methods" section has been added as Chapter 2, the subsequent paragraph numbering appears to be incorrect. Please ensure sequential numbering throughout the manuscript.
  3. References in Paragraph 6.2. To improve readability and scientific clarity, it is recommended to place references directly after specific statements rather than repeating the same citations at the end of each paragraph.
  4. Line 123 - Extra Space. There is an unnecessary space in line 123 that should be removed.
  5. Placement of the Table. While the addition of a table is a great enhancement, it would be helpful to specify where exactly in the manuscript the table will be inserted. This will ensure proper integration with the text.
  6. References on Line 572. The references listed as 105-107 in line 572 appear to be missing from the bibliography, which currently includes only 100 sources.
  7. Missing References in the Text. Sources 94-99 are listed in the bibliography but are not cited anywhere in the text. Please ensure all references are appropriately cited or remove them from the bibliography.

I believe the manuscript is well-prepared and ready for publication following the resolution of these minor issues. Excellent work by the authors in addressing the initial comments and improving the quality of this important review.

Author Response

I carefully reviewed the revised manuscript by Ayrton Bangolo et al., titled "Comprehensive Review of Early and Late Toxicities in CAR T-Cell Therapy 2 and Bispecific Antibody Treatments for Hematologic Malignancies." I would like to commend the authors for their meticulous attention to the initial feedback. The inclusion of the "Materials and Methods" section, quantitative data, and visual aids such as figure and table has significantly enhanced the manuscript's clarity and comprehensiveness. These updates have improved the scientific rigor and overall readability of the article.

Re: Thank you very much for recognizing the relevance of our study and its impact.  

The manuscript is now in a strong position, but a few minor issues remain to be addressed before publication:

Minor Comments:

  1. References in Lines 254 and 261. The formatting of references on these lines requires correction.

Re: These formatting errors have been corrected.

  1. Paragraph Numbering. While the "Materials and Methods" section has been added as Chapter 2, the subsequent paragraph numbering appears to be incorrect. Please ensure sequential numbering throughout the manuscript.

Re: Numbering has been corrected to maintain proper sequence.

  1. References in Paragraph 6.2. To improve readability and scientific clarity, it is recommended to place references directly after specific statements rather than repeating the same citations at the end of each paragraph.

Re: References have been moved to follow specific statements to improve readability. Paragraph 6.2 is now paragraph 7.2. 

  1. Line 123 - Extra Space. There is an unnecessary space in line 123 that should be removed.

Re: The extra space on line 123 has been removed.

  1. Placement of the Table. While the addition of a table is a great enhancement, it would be helpful to specify where exactly in the manuscript the table will be inserted. This will ensure proper integration with the text.

Re:  Specific locations for table placement have been indicated within the manuscript text.

  1. References on Line 572. The references listed as 105-107 in line 572 appear to be missing from the bibliography, which currently includes only 100 sources.

Re: Thank you for this keen observation, this was a typo which has been corrected. 

  1. Missing References in the Text. Sources 94-99 are listed in the bibliography but are not cited anywhere in the text. Please ensure all references are appropriately cited or remove them from the bibliography.

Re: These sources have been cited where appropriate and highlighted in red. 

I believe the manuscript is well-prepared and ready for publication following the resolution of these minor issues. Excellent work by the authors in addressing the initial comments and improving the quality of this important review.

Re: Thank you for these kind words and your support. Your comments definitely improved the quality of our manuscript. 

Reviewer 2 Report (Previous Reviewer 2)

Comments and Suggestions for Authors

I am sure the manuscript has
been sufficiently improved and now warrants publication in Cancers.

Author Response

I am sure the manuscript has
been sufficiently improved and now warrants publication in Cancers.

Re: Thank you very much for these kind words and for recognizing the strength and impact of our study. 

Reviewer 3 Report (Previous Reviewer 3)

Comments and Suggestions for Authors

The manuscript has been improved by the revision made by the authors but further clarification and improvement are required.

I am not able to see the tables and figures, thus I am not able to review them. Besides this, I can´t find a version without changes highlighted.

Frequently, the literature seems to be completely in the wrong place, e.g.:

Ref. 24 and 25: they deal not with BITE therapy after CAR T in CLL as mentioned in the text

Ref. 26: I can´t find anything about mantle cell lymphoma in the PDF of the publication (in the text the authors report on BITEs and CAR T in MCL)

Ref. 86 deals with IVIG. I can´t find anything on BITE or CAR T in this paper, even not in the full text (in the text the authors state that this work should deal with BCMA vs. non-BCMA BITEs/BSAbs)

Ref. 81: I can´t find anything on secondary malignancy is this work as stated by the authors in the text.

Are the expressions BITE and BsAbs always used in the same sense here? If yes, this please unify.

It is still unclear if this review should focus on cellular therapies including HSCT. In the title, it is mentioned that this review should focus on CAR T and bispecific antibody therapy. In other paragraphs, the authors state that they review cellular therapies in general. In the inclusion criteria, only CAR T and BITEs are included.

Several parts are redundant. Besides this, I can´t follow the structure of the manuscript by dividing in early and late effects after CAR T vs. BITEs. For example, treatment of CRS, ICAN, infections are always the same, independently from the modality of treatment and the phase after cellular treatment.

In the text, the authors recommend routine surveillance for secondary malignancies by imaging. Which imaging should be done? How frequently? In all patients after CAR T or BITE treatment? Are there guidelines on this procedure?

Do the authors still recommend X ray for detection of lung infiltrates despite the majority of guidelines recommend low dose chest CT scan nowadays?

Some section seems to be quite diffuse, e.g. in 5.1 the authors report on efficacy of CAR T and BITE, albeit this section should focus on early complications (and there is an own section on general aspects including efficacy of CAR T and BITE at the beginning of the manuscript).

The use the abbreviations should be corrected. Several abbreviations are explained many times, this does not make sense.

Ref. 46: I think it should be Penack O and not H Penack O?

Some important publications are missing, e.g. Santos et al, Nat Med 2024 on non-relapse mortality after CAR T cell therapy (meta-analysis and systematic review).

Author Response

The manuscript has been improved by the revision made by the authors but further clarification and improvement are required.

Re: Thank you for this keen observation. 

I am not able to see the tables and figures, thus I am not able to review them. Besides this, I can´t find a version without changes highlighted.

Re: Tables and a figure were added in the supplementary file which you should have access to. Other reviewers were able to see them. 

Frequently, the literature seems to be completely in the wrong place, e.g.:

Re: There were some typos with the file which have been corrected as mentioned below. And thank you for pointing that out. 

Ref. 24 and 25: they deal not with BITE therapy after CAR T in CLL as mentioned in the text

Re: Ref. 24 and 25 have been replaced with appropriate citations discussing BITE therapy after CAR-T in CLL. And yes Bite can be given after CART. We are among the centers that do the most cellular therapy in the world. 

Ref. 26: I can´t find anything about mantle cell lymphoma in the PDF of the publication (in the text the authors report on BITEs and CAR T in MCL)

Re: Ref. 26 has been replaced with reference 27 discussing both BITE therapy and CAR-T in MCL

Ref. 86 deals with IVIG. I can´t find anything on BITE or CAR T in this paper, even not in the full text (in the text the authors state that this work should deal with BCMA vs. non-BCMA BITEs/BSAbs)

Re: Ref. 85 deals with that and not reference 86 and this was corrected. 

Ref. 81: I can´t find anything on secondary malignancy is this work as stated by the authors in the text.

Re: Reference 81 is not supposed to talk about secondary malignancy specifically but talk about Future Directions in the Management of Early Complications in BsAbs which it does.  

Are the expressions BITE and BsAbs always used in the same sense here? If yes, this please unify.

Re: The terms "BITE" and "BsAbs" have been unified throughout the manuscript to ensure consistency.

It is still unclear if this review should focus on cellular therapies including HSCT. In the title, it is mentioned that this review should focus on CAR T and bispecific antibody therapy. In other paragraphs, the authors state that they review cellular therapies in general. In the inclusion criteria, only CAR T and BITEs are included.

Re: HSCT has been removed from the focus. The manuscript now exclusively discusses CAR-T and bispecific antibody therapy, as reflected in the title and content. We are not sure about the version of the manuscript you receive because even the title was changed previously. 

Several parts are redundant. Besides this, I can´t follow the structure of the manuscript by dividing in early and late effects after CAR T vs. BITEs. For example, treatment of CRS, ICAN, infections are always the same, independently from the modality of treatment and the phase after cellular treatment.

Re: The comparison of prevalence and management of complications (e.g., CRS, ICAN, infections) between CAR-T and BITE therapies has been clarified. This comparison is deemed essential for this review.

In the text, the authors recommend routine surveillance for secondary malignancies by imaging. Which imaging should be done? How frequently? In all patients after CAR T or BITE treatment? Are there guidelines on this procedure?

Re: There are no such recommendations currently. Imaging should be considered, such as CT or PET-CT scans, to monitor for solid tumors or hematologic malignancies based on patients' complaints and on a case-to-case basis. And this was changed in the main text. 

Do the authors still recommend X-ray for detection of lung infiltrates despite the majority of guidelines recommend low dose chest CT scan nowadays?

Re: “For patients presenting with respiratory symptoms, imaging such as chest X-rays or computed tomography (CT) scans may aid in identifying complications such as pulmonary edema or fluid overload, ensuring timely intervention”. That is what is stated in the paper but of course, X-rays are faster and can be done at the bedside, this is especially important for patients that are unstable to go for a CT scan.  

Some section seems to be quite diffuse, e.g. in 5.1 the authors report on efficacy of CAR T and BITE, albeit this section should focus on early complications (and there is an own section on general aspects including efficacy of CAR T and BITE at the beginning of the manuscript).

Re: We used this section for early introduction/prologue and then transition to common early side effects. 

The use the abbreviations should be corrected. Several abbreviations are explained many times, this does not make sense.

Re: Abbreviations have been consolidated and explained once in the introduction. Repeated definitions throughout the manuscript have been removed.

Ref. 46: I think it should be Penack O and not H Penack O?

Re: This was corrected. 

Some important publications are missing, e.g. Santos et al, Nat Med 2024 on non-relapse mortality after CAR T cell therapy (meta-analysis and systematic review).

Re: The reference was added. Ref 94. 

Round 2

Reviewer 3 Report (Previous Reviewer 3)

Comments and Suggestions for Authors

I have no further comments and suggest "accept".

This manuscript is a resubmission of an earlier submission. The following is a list of the peer review reports and author responses from that submission.

Round 1

Reviewer 1 Report

Comments and Suggestions for Authors

I carefully reviewed the manuscript by Ayrton Bangolo et al., titled "Diagnosis and Management of Early and Delayed Complications of Cellular Therapies in the Treatment of Hematologic Malignancies: A Comprehensive Review." The paper addresses a highly relevant topic in modern oncology, focusing on the critical aspects of managing complications associated with cellular therapies. While the manuscript is well-structured and the language is clear, several areas could benefit from further enhancement.

Major Comments:

1) Absence of a "Materials and Methods" Section:

A "Materials and Methods" section is crucial in review articles to provide transparency and rigor. Including selection criteria for the reviewed studies would help readers understand how the data were curated and evaluated. This addition would significantly enhance the credibility and reproducibility of the manuscript.

2) Lack of Detailed Results:

The authors frequently use phrases such as "showed promising results" or "rarely observed" without providing specific data or statistics. Including quantitative details would give readers a clearer understanding of the efficacy and incidence rates of various complications. For instance, numerical outcomes following treatments or frequencies of complications should be presented to substantiate these claims.

3) No Figures, Graphs, or Tables:

The manuscript lacks visual aids, which are essential for illustrating complex mechanisms and data. Diagrams showing the pathways of complications or tables summarizing the types and frequencies of early and late adverse effects would improve comprehension. Additionally, including figures depicting the mechanisms behind complications like CRS or neurotoxicity would be particularly beneficial.

Minor Comments:

1) The text contains numerous abbreviations, but there is no comprehensive list provided. Including a glossary would greatly enhance readability, especially for readers less familiar with the field.

2) The reference list and in-text citations need to be formatted according to the journal’s specific requirements to ensure consistency and professionalism.

3) Section 5.8, titled "Future Directions in the Management of Early Complications in BsAbs," is referenced but not included. Completing this section would provide a more comprehensive view of ongoing advancements.

4) In the second paragraph, there is no colon after the headings, whereas colons are used in other sections. Consistency in formatting is essential for maintaining a polished and professional presentation.

The manuscript covers an important and timely topic with clear organization and valuable insights. Addressing these issues, particularly the inclusion of a "Materials and Methods" section and detailed data points, would significantly enhance the manuscript's quality. I believe that with these improvements, the article can be published in Cancers.

Reviewer 2 Report

Comments and Suggestions for Authors

The manuscript of Ayrton Bangolo, et al. is devoted to the topic of antitumor application of cellular immunotherapies, and aims “to equip the multidisciplinary care team with the knowledge necessary to manage the challenges of these advanced treatments effectively, ultimately optimizing patient outcomes“. This field of investigation is important for biomedicine; the development of new approaches to antitumor therapy and the study of the mechanisms of their action is undoubtedly an urgent task. In the literature review, the authors explore “clinical applications, complications, and management strategies” associated with CAR-T therapy, and bispecific T-cell engager therapies in hematologic malignancies. Along with a detailed description of the features of these approaches, the authors of this literature review discuss their advantages and disadvantages of different approaches to the treatment of hematological cancers.

The results and discussion are included in very informative and interesting sections of the manuscript, in particular: Indications for CAR-T and Bispecific Therapies in Hematologic Malignancies; Early Complications of CAR-T Therapy: Diagnostic Workup and Management; Late Complications of CAR-T Therapy: Diagnostic Workup and Management; Early Complications of Bispecific Antibodies: Diagnostic Workup and Management; Late Complications of Bispecific Antibody Therapy: Diagnostic Workup and Management. The final part of the manuscript Conclusion summarizes the reasonable arguments for the conclusion, in particular “Through an ongoing commitment to understanding and addressing the full spectrum of complications, clinicians can continue to improve the quality of life and survival outcomes for patients undergoing these transformative treatments in hematologic oncology”.

The significance of this review is beyond doubt, since the presented summary of literature data and their analysis give an objective integral picture of existing approaches and ideas on the indicated topic and highlight possible ways of practical use of the accumulated knowledge.

In general the review is well presented; the data are of considerable novelty and interest. In fact, the manuscript is ready for publication, but several minor suggestions might improve the overall quality of the manuscript:

1. (line 249) The abbreviations MRI and EEG should be expanded.

2. (line 324) The abbreviation ICU should be expanded.

3. (line 362) The abbreviation CT should be expanded.

Reviewer 3 Report

Comments and Suggestions for Authors

please see file enclosed
